# Dysbiosis in the Nasal Mycobiome of Infants Born in the Aftermath of Hurricane Maria

**DOI:** 10.3390/microorganisms13081784

**Published:** 2025-07-31

**Authors:** Ruochen Wang, David de Ángel Solá, Félix E. Rivera-Mariani, Benjamín Bolaños Rosero, Nicolás Rosario Matos, Leyao Wang

**Affiliations:** 1Department of Biostatistics and Epidemiology, School of Public Health and Health Sciences, University of Massachusetts, Amherst, MA 01003, USA; ruochenwang@umass.edu; 2San Juan City Hospital, San Juan 00921, Puerto Rico; david.deangelss@gmail.com (D.d.Á.S.); nrosario@criggroup.com (N.R.M.); 3College of Arts and Sciences, Lynn University, Boca Raton, FL 33431, USA; friveramariani@lynn.edu; 4RIPLRT Institute, Fort Lauderdale, FL 33301, USA; 5Department of Microbiology and Immunology, School of Medicine, University of Puerto Rico, San Juan 00925, Puerto Rico; benjamin.bolanos@upr.edu; 6Institute for Applied Life Sciences (IALS), University of Massachusetts, Amherst, MA 01003, USA

**Keywords:** climate change, infant nasal mycobiome, Hurricane Maria, fungi, HOLA cohort study

## Abstract

Hurricanes and flooding events substantially elevate indoor fungal spore levels, which have been associated with increased risks of developing childhood asthma and other adverse respiratory outcomes. Although environmental fungal compositions following major hurricanes have been well characterized, the fungal communities within the nasal cavity (i.e., the nasal mycobiome) of exposed individuals remain unexplored. We collected nasal swab samples from infants following Hurricane Maria in San Juan, Puerto Rico, during two periods (March to August 2018 and February to September 2019). We processed a total of 58 samples (26 from the first year and 32 from the second year post-Hurricane Maria) and performed internally transcribed spacer (ITS) rRNA gene sequencing to characterize and compare the infant nasal mycobiome between the two groups. Although alpha-diversity did not differ significantly, beta-diversity analyses revealed significantly different fungal compositions between the two groups (*p* <0.01). Infants exposed during the first year post-Hurricane Maria had significantly higher abundances of *Alternaria*, *Eutypella*, *Schizophyllum*, and *Auricularia,* compared to infants from the second year. *Alternaria* was also more prevalent in the first-year than in the second-year infants (42% vs. 9%, *p* = 0.01). Our study provides evidence linking early-life hurricane exposures to elevated risks of developing childhood asthma.

## 1. Introduction

Hurricanes and extensive flooding events create environments that foster mold growth, posing profound health risks [1,2,3,4,5,6]. Studies on Hurricane Katrina and other major flooding events have repeatedly documented that fungal species, including *Alternaria*, *Aspergillus*, *Penicillium*, and *Cladosporium*, were dominant in flooded houses [4,7,8,9,10]. These remarkably elevated levels of mold spores led to shifts in fungal community compositions. A study on Hurricane Maria reported an altered indoor fungal community that persisted for over a year, with recovery only occurring after almost two years post-hurricane [5]. Exposure to the post-hurricane environment had direct impacts on allergic symptoms and asthma exacerbations [11]. In addition, the incidence of invasive mold infections was reported to be 1.48 times higher and more severe within one year after Hurricane Harvey compared to one year before the hurricane [6]. These studies highlighted fungal spores as a key environmental factor in mediating disease risks after hurricanes.

Fungal spores that spike in the aftermath of hurricanes and flooding events have lasting effects on respiratory health in children [4,12,13,14]. In a prospective birth cohort study, early-life exposure to high levels of *Alternaria*, *Cladosporium*, and *Penicillium* was associated with an elevated risk of developing lower respiratory infections and wheezing during the first year of life [6]. Indeed, exposure to moldy indoor environments during infancy has been shown to remarkably increase the risk of developing childhood asthma [15,16]. These findings suggest that the inhalation and colonization of fungal spores in the infant airway elicited long-term implications for the development of asthma. Therefore, it is essential to characterize the fungal communities in the infant upper airway (i.e., the infant nasal mycobiome) after being exposed to a major hurricane. However, to our knowledge, no such study has attempted to describe the infant nasal mycobiome in a post-hurricane setting. This critical knowledge gap limits our comprehensive understanding of the relationship between hurricane exposures during early infancy and subsequent asthma risks during childhood, thereby hindering our efforts to develop effective screening and intervention strategies for childhood asthma in the era of climate change.

To this end, we performed our Hurricane as the Origin of Later Alterations in Microbiome (HOLA) cohort study in San Juan, Puerto Rico, following Hurricane Maria, a Category 4 storm that struck the island in 2017. With robust sequencing and survey data, we previously characterized the infant microbiome features associated with early life hurricane exposures [17,18,19]. In this study, we utilized the collected infant nasal swab samples to perform internally transcribed spacer (ITS) rRNA gene sequencing and characterize the nasal mycobiome of hurricane-exposed infants. We compared the nasal microbiome of infants born within the first year following Hurricane Maria (sample collection period: March to August 2018) to those born two years after the hurricane (sample collection period: February to September 2019).

This study aimed to evaluate differences in the nasal mycobiome profile of infants exposed to a major hurricane during the first year versus the second year post-hurricane. We hypothesized that infants exposed during the first-year post-hurricane would harbor an altered nasal mycobiome with a higher abundance of dampness-associated fungal taxa compared to those born one year later.

## 2. Materials and Methods

### 2.1. Cohort Recruitment

The HOLA cohort study was conducted after receiving approval from the Institutional Review Board of San Juan City Hospital (ID: EPDC-Microbiome. Approval Date: 15 March 2018). To compare the nasal mycobiome of infants born within one year after Hurricane Maria to those born one to two years after the hurricane, we recruited infants between March and August 2018 (Year 1 Group) and then between February and September 2019 (Year 2 Group) at the San Juan City Hospital Research Unit. Written informed consent was obtained from all participating parents. Eligibility criteria included full-term infants (≥37 weeks of gestational age at the time of birth) who were delivered vaginally, aged between two to six months, and whose mothers had resided in Puerto Rico throughout their pregnancy. Exclusion criteria included delivery by cesarean section, premature birth (<37 weeks of gestational age at the time of birth), admission to a neonatal intensive care unit, acute illness at the time of sampling, airway or pulmonary malformations, and identified chromosomal or genetic abnormalities. During the postpartum clinic visit, each infant’s characteristics were collected through a questionnaire completed by the mother.

### 2.2. Sample Collection, Processing, and Sequencing

During the postpartum clinic visits, a trained health professional collected nasal swab samples by inserting a sterile swab into both nostrils of each infant against the nasal mucosa. Each swab was immediately placed into a DNA/RNA Shield collection tube (Zymo Research, Irvine, CA, USA) containing a stabilization solution that preserved the integrity of nucleic acids at room temperature. All samples were shipped to the Wang Lab at Washington University in St. Louis for processing. The entire microbial DNA was extracted from each collected sample using the ZymoBIOMICS DNA Miniprep Kit (Zymo Research). The library for ITS rRNA gene sequencing was generated using the Quick-ITS Plus NGS Library Prep Kit (Zymo Research). Labeled samples were pooled and sequenced on the Illumina MiSeq platform with 250 bp paired-end reads at the Washington University in St. Louis DNA Sequencing Innovation Lab.

### 2.3. Sequencing Data Analysis

The sequencing reads were analyzed using the DADA2 pipeline to ascertain sequence variants [20], and taxonomic classification was conducted using the UNITE database (version 04.02.2020) [21]. Samples yielding less than 100 reads after quality filtering were excluded from further compositional analyses to ensure data quality. The top 20 most abundant genera across samples were identified by ranking their relative abundance. The alpha-diversity, including the number of observed taxa and Shannon index, was evaluated between the two different groups. The Wilcoxon rank-sum test was used to assess statistical significance for alpha-diversity. The beta-diversity (e.g., the principal coordinates analysis [PCoA]) was based on Bray-Curtis distances, while the significance was evaluated using analysis of similarity (ANOSIM). Differentially abundant fungal genera between the two groups were identified using the DESeq2 algorithm [22]. Only genera present in at least 20% of all the samples were included. The *p*-values were adjusted for multiple comparisons using the Benjamini-Hochberg method. The prevalence of *Alternaria* and other selected genera between the Year 1 and Year 2 Groups was compared using the Chi-square test or Fisher’s exact test. An alpha level less than 0.05 was considered statistically significant. All statistical analyses were performed using R version 4.4.1.

## 3. Results

### 3.1. The HOLA Cohort Study and Profiling of the Infant Nasal Mycobiome

We successfully characterized 26 and 32 nasal mycobiomes from the Year 1 and Year 2 Groups, respectively. All participating infants and their families lived in the San Juan metropolitan area. Figure 1 shows the municipalities in which the infants from the Year 1 and Year 2 Groups were located. Table 1 compares the basic characteristics of the two groups.

The ITS rRNA gene sequencing yielded a total of 414,031 high-quality reads from the DADA2 pipeline. The Year 1 Group (*n* = 26 infants) was comprised of 184,634 reads, and the Year 2 group (*n* = 32 infants) accounted for 229,397 high-quality reads. Across all 58 nasal swab samples, a total of 1093 amplicon sequence variants (ASVs) representing unique fungal strains were identified. These ASVs belong to 343 different genera. The top 20 most abundant fungal genera across the 58 samples were *Malassezia*, *Rigidoporus*, *Starmerella*, *Scopuloides*, *Yarrowia*, *Candida*, *Clavispora*, *Trametes*, *Cladosporium*, *Alternaria*, *Nigrospora*, *Aspergillus*, *Xylodon*, *Resinicium*, *Eutypella*, *Sistotremastrum*, *Trichosporon*, *Curvularia*, *Leptospora*, and *Bjerkandera*. Figure 2A displays the relative abundance of these genera in each nasal sample by group.

### 3.2. Mycobiome Comparisons at the Community Level

We then compared the infant nasal mycobiomes between the Year 1 and Year 2 Groups. To assess alpha-diversity, which reflects the diversity of species within specific samples, we compared the number of observed unique ASVs (to measure richness in each sample) as well as the Shannon index (to measure richness and evenness in each sample) between the two groups. As shown in Figure 2B, the number of observed ASVs was not significantly different between the two groups (*p*-value = 0.14). Likewise, the Shannon index did not show significant differences between the two groups either (*p*-value = 0.27). We then evaluated beta-diversity, which measured dissimilarities in compositional differences between two mycobial communities. As shown in Figure 2C, the Year 1 and Year 2 Groups had significantly different mycobiome compositions (*p*-value < 0.01), meaning infants in the Year 1 Group harbored a very different fungal community in their nasal cavity compared to infants in the Year 2 Group.

In the Year 1 Group, mothers were asked about their hurricane-related exposures during pregnancy, and eight mothers reported being exposed to mold. Therefore, we evaluated whether self-reported maternal mold exposure was an important factor in shaping the infant nasal mycobiome at the community level. There were no significant differences regarding the alpha-diversity (*p*-value = 0.13 for number of observed taxa analysis, and *p*-value = 0.76 for Shannon Index analysis, Figure 2D) or beta-diversity (*p*-value = 0.57, Figure 2E) by self-reported maternal mold exposure in the Year 1 Group.

We evaluated potential influencing variables, including age, sex, breastfeeding status, and sibling presence at home, on the infant nasal mycobiome across the entire cohort including both Year 1 and Year 2 group, but we did not detect significant impacts from these factors (Appendix A).

### 3.3. Mycobiome Comparisons at the Genus Level

Next, we performed analyses using the DESeq2 algorithm to identify differentially abundant fungal genera between the two groups. Four fungal genera (*Alternaria*, *Eutypella*, *Auricularia*, and *Schizophyllum*) were found to be significantly more abundant in the Year 1 Group compared to the Year 2 Group (Figure 3A). Importantly, *Alternaria* is a fungal genus that has been reported to be associated with house dampness and has been recognized as an asthma-associated allergen. Therefore, we further examined the prevalence of *Alternaria* in each group and found that *Alternaria* was significantly more prevalent among the infants from the Year 1 Group than in the Year 2 Group (42% vs. 9%, *p*-value = 0.01, Figure 3B).

In addition to *Alternaria*, several other fungal genera have also been repeatedly reported to be increased in a post-hurricane or flooding environment, including *Aspergillus*, *Cladosporium*, and *Penicillium*. However, we did not detect any significant differences between the two groups based on the DeSeq2 algorithm. To confirm that these fungal genera were not differentially abundant between the two groups, we performed analyses focusing on these three fungal genera. As shown in Figure 4A,B, there were no significant differences regarding their relative abundances or prevalence between the Year 1 and Year 2 Groups. However, four other fungal genera (*Lentinus*, *Trametes*, *Yarrowia*, and *Nigrospora*) were found to be significantly more abundant in the Year 2 Group compared to the Year 1 Group (Figure 3A).

## 4. Discussion

Climate change has increased the frequency and intensity of extreme weather events, such as hurricanes and flooding events [23]. These climate extremes pose significant risks to children’s respiratory health [24], with adverse respiratory outcomes believed to be mainly attributed to environmental fungal exposures. However, studies evaluating fungal communities in the airways of exposed populations have been lacking, as prior research has predominantly focused on characterizing environmental fungal communities following hurricanes. Addressing this critical knowledge gap will allow us to better link environmental fungal changes with human respiratory health.

Our HOLA cohort study, established in San Juan, Puerto Rico, following Hurricane Maria, begins to fill this important knowledge gap. Our results revealed that infants recruited during the first year after Hurricane Maria had a significantly different nasal mycobiome compared to infants recruited one year later. Infants exposed during the first year after Hurricane Maria harbored a higher abundance and prevalence of *Alternaria,* a hurricane- and dampness-associated fungal genus, in their nasal cavity than those born one year later. These results suggested that fungal communities in the infant upper airway may have been influenced by environmental mold after Hurricane Maria. Therefore, our study provided valuable evidence on the influential effects of environmental mold spores on fungal communities in the infant’s upper airway. This influential effect may play an important role in mediating the association between hurricane exposures during infancy and increased risks of developing childhood asthma.

*Alternaria* is a prevalent indoor mold allergen, and its growth has been associated with post-hurricane water damage [25,26]. Interestingly, the Head-off Environmental Asthma in Louisiana (HEAL) study reported that *Alternaria* was the only allergen still found at high prevalence in house dust samples collected in New Orleans 19 to 35 months post-Hurricane Katrina, even after mold remediation and renovations [26]. This finding indicated the persistent presence of *Alternaria*, but not other genera, in indoor post-hurricane settings. Consistent with this, our results showed that among several fungal genera that have been repeatedly reported to be associated with hurricane and flooding events, only *Alternaria* was found to be remarkably enriched in the infant nasal swab samples collected one year after Hurricane Maria. Notably, *Alternaria* exposures have been closely associated with childhood asthma development and severe asthma exacerbations through airway epithelial barrier disruption and immune activation [12,27,28,29,30]. Therefore, *Alternaria* may be an essential pathway through which post-hurricane environments may lead to pediatric asthma. Our study emphasized the importance of examining fungal contaminants, particularly *Alternaria,* in the indoor environment after hurricanes.

Besides *Alternaria*, we also identified three other genera, including *Schizophyllum, Eutypella*, and *Auricularia*, that were enriched in the Year 1 Group and are of particular interest due to their potential relevance to adverse respiratory health outcomes. *Schizophyllum* and *Eutypella* have been related to respiratory disease in previous studies. For instance, *Schizophyllum*, an emerging allergen and pathogen, can enhance both the severity of asthma and the frequency of exacerbations [31]. In addition, inhalation of *Schizophyllum* can cause allergic bronchopulmonary mycosis [32]. *Eutypella* has been found to be enriched in the sputum mycobiome of patients with chronic obstructive pulmonary disease (COPD) [33]. To our knowledge, no studies have reported an association between *Auricularia* and lung-related diseases. However, *Auricularia* has been found in the gut mycobiome and has been associated with gestational diabetes mellitus. It also remains unclear whether enriched *Auricularia* in the infant nasal mycobiome has any effects on an infant’s health. It is worth mentioning that we also identified four other fungal genera, including *Lentinus*, *Trametes*, *Yarrowia*, and *Nigrospora*, that were more enriched in the Year 2 Group compared to the Year 1 Group. However, none of these four fungal genera were reported to be associated with hurricanes, flooding events, or asthma.

Several limitations in this study should be noted. First, we did not collect environmental samples, so we were unable to evaluate the relationship between environmental fungi exposures and the infant nasal mycobiome. As emerging evidence has supported an influential role of the environment on the human mycobiome, [34] future studies that collect both human and environmental samples after hurricanes will be important to validate environmental influences on the infant nasal mycobiome. Second, prior studies have shown that exposure to indoor mold and dampness during the prenatal stage is associated with increased risks of developing asthma [35]. It is possible that prenatal hurricane exposures played a role in mycobiome dysbiosis in the Year 1 Group since all the infants in that group were exposed in utero to Hurricane Maria. However, given the study design, we cannot distinguish the prenatal and postnatal effects on the infant nasal mycobiome. Third, we did not follow the participants to assess their respiratory outcomes, such as wheezing, viral respiratory infections, and childhood asthma. Studies that integrate airway mycobiome profiling and respiratory health outcomes are needed to further evaluate the associations between nasal fungal dysbiosis during infancy and the risks of developing respiratory diseases. Fourth, we used a cohort of infants who were born between one and two years after Hurricane Maria as the control group to identify hurricane-related dysbiosis. It is important to note, however, that the profound impacts of the devastating Hurricane Maria were long-lasting and could have affected the control group of infants as well. Lastly, the relatively small sample size of our HOLA cohort may have limited our power to detect important fungal taxa that could be relevant among hurricane-exposed infants.

## 5. Conclusions

In summary, we conducted the HOLA cohort study, which recruited infants one and two years following Hurricane Maria in the metropolitan area of San Juan, Puerto Rico. Our study revealed that the infant nasal mycobiomes from the two groups had significantly different compositions. More importantly, infants born during the first-year post-hurricane harbored more hurricane- and asthma-related taxa, especially *Alternaria*, in their nasal cavity compared to those born during the second year post-hurricane. Our study suggests that environmental fungal allergens may directly colonize the infant’s nasal cavity, potentially increasing the risk of childhood asthma and other respiratory diseases. This study moved beyond environmental fungal measurements by evaluating the human mycobiome in the context of hurricanes and extreme weather conditions.

## Figures and Tables

**Figure 1 microorganisms-13-01784-f001:**
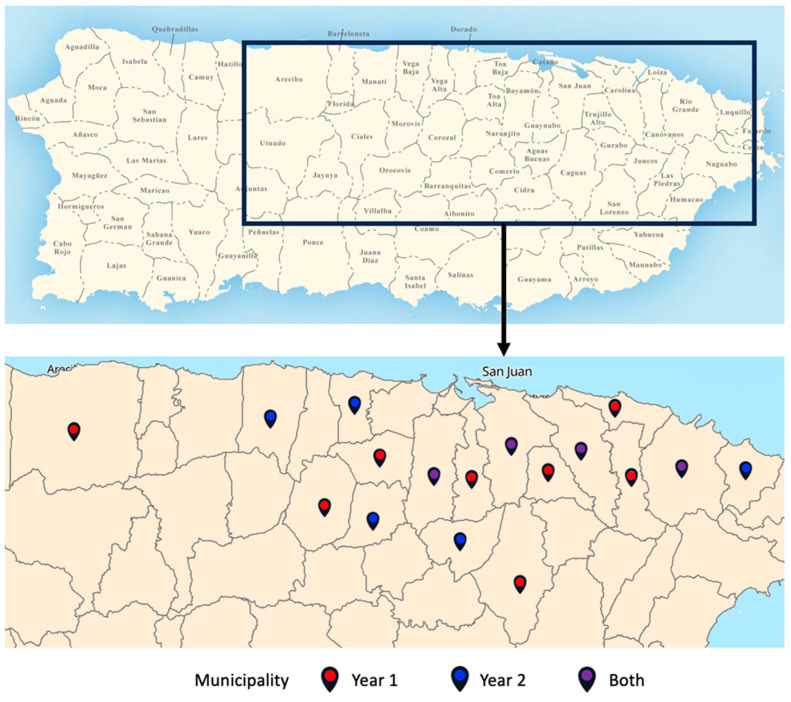
Map showing the residential locations of study participants in Puerto Rico.

**Figure 2 microorganisms-13-01784-f002:**
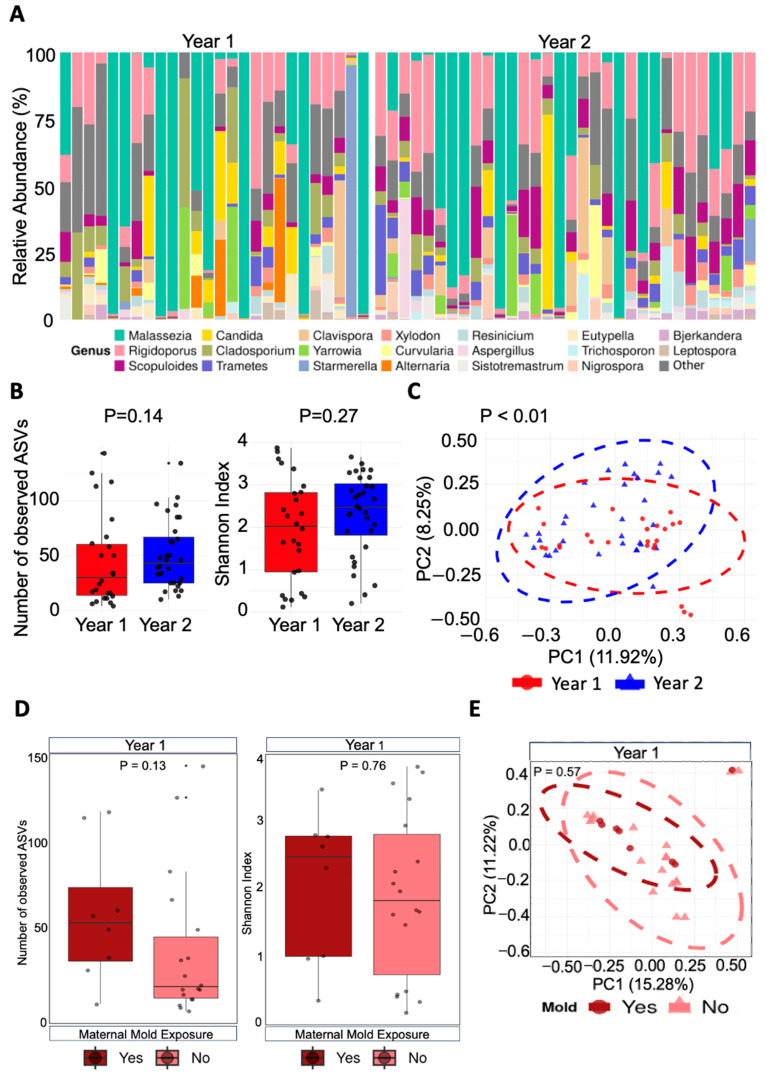
Comparison of the infant nasal mycobiome at the community level. (**A**) The nasal mycobiome with each bar representing a nasal sample. The top 20 abundant fungal genera are displayed. (**B**) Alpha-diversity analyses between both groups. Statistical significance was based on the Wilcoxon rank-sum test. (**C**) Principal coordinate analysis (PCoA) plot was based on the Bray-Curtis distance. Statistical significance was based on the analysis of similarity (ANOSIM) test. (**D**) Alpha-diversity analyses by self-reported fungal exposure in the Year 1 Group. Statistical significance was assessed using the Wilcoxon rank-sum test. (**E**) PCoA plot based on the Bray-Curtis distance by fungal exposure in Year 1 Group. Statistical significance was analyzed with the ANOSIM test.

**Figure 3 microorganisms-13-01784-f003:**
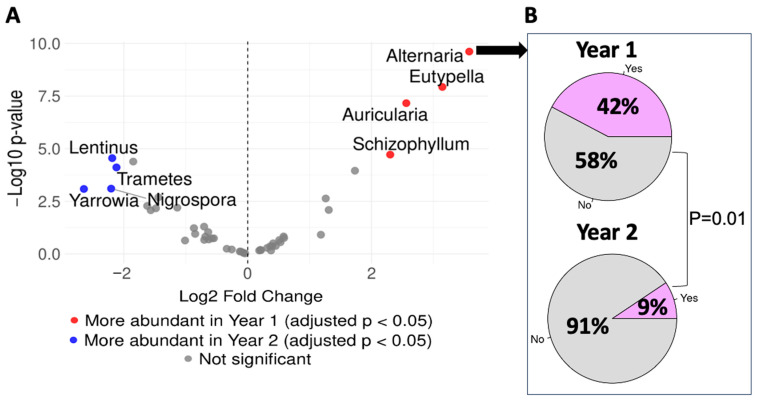
*Alternaria* was more abundant and prevalent in the Year 1 Group than the Year 2 Group. (**A**) Volcano plot displaying differentially abundant fungal genera. Data were analyzed using the DESeq2 algorithm (excluding genera present in <20% of samples) and adjusted for multiple comparisons using the Benjamini-Hochberg method. Red and blue dots indicate significantly abundant genera. (**B**) *Alternaria* prevalence is shown by group. Statistical significance was assessed using the Chi-Square test.

**Figure 4 microorganisms-13-01784-f004:**
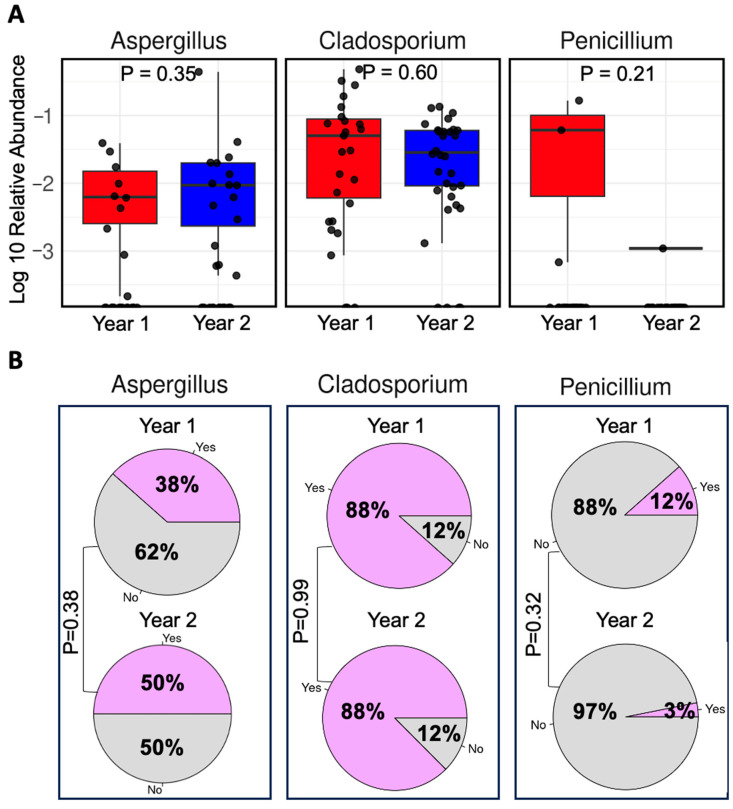
Comparison of selected genera. (**A**) Relative abundance of selected genera between the two groups. Statistical significance was assessed using the Wilcoxon rank-sum test. (**B**) Prevalence of selected genera by group. Statistical significance was evaluated using the Chi-square test for *Aspergillus* and Fisher’s exact test for *Cladosporium* and *Penicillium*.

**Table 1 microorganisms-13-01784-t001:** Characteristics of the study participants.

Variables.	Year 1 Group (*n* = 26)	Year 2 Group (*n* = 32)	*p*-Value
Male sex: no. (%)	9 (34.62)	18 (56.25)	0.17
Age (wk), mean (SD)	16.10 (5.51)	16.17 (4.44)	0.81
Gestational age (wk), mean (SD)	38.84 (0.94)	38.85 (1.40)	0.96
Sibling presence at home, no. (%)	19 (73.08)	22 (68.75)	0.94
Exclusive breastfeeding, no. (%)	7 (26.92)	8 (25.00)	0.99
Maternal exposure to mold, no. (%)	8 (30.77)	0 (0)	—

Note: Statistical significance was evaluated using the Wilcoxon rank-sum test or Chi-square test. SD: standard deviation.

## Data Availability

The raw sequencing reads have been archived in the European Nucleotide Archive (BioProject ID: PRJNA1258278).

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
