# Peer review of "Dysbiosis in the Nasal Mycobiome of Infants Born in the Aftermath of Hurricane Maria"

_microorganisms, 2025, doi:10.3390/microorganisms13081784_

Round 1
Reviewer 1 Report
Comments and Suggestions for Authors
Wang and colleagues present an interesting and well-written cross-sectional study on the short-term effects of exposure to flooding/hurricanes on the nasal mycobiome in Puerto Rican infants. The study strictly investigates the mycobiome of those born within a year of Hurricane Maria vs those around 1 year after that. This information may be clear in the introduction and aims, but should be made more clear in the 2.1 subsection of the methods. The manuscript is otherwise easy to read and follow. However, the authors do not attempt to establish associations between the nasal mycobiome and important respiratory outcomes, which would certainly be feasible, since at the age of 6-7 (at which the children should be in the present day) it would be possible to determine at least if these children are atopic/allergic, and have/have had recurrent wheezing/asthma. This would greatly improve the quality and reach of the manuscript.
A few further comments:
Line 74: Aims should be in a new paragraph.
Lines 130-137: this sentenced is redundant and does not add information besides what is already clear in table 1, and should be removed.
Line 253-4: do the authors mean have been related to respiratory diseases?
Line 256: Schizophyllum should be in italic
Limitations should include the absense of a control group from a non-flooded area.
Author Response
Comment 1: Wang and colleagues present an interesting and well-written cross-sectional study on the short-term effects of exposure to flooding/hurricanes on the nasal mycobiome in Puerto Rican infants. The study strictly investigates the mycobiome of those born within a year of Hurricane Maria vs those around 1 year after that. This information may be clear in the introduction and aims, but should be made more clear in the 2.1 subsection of the methods.
Response 1: We thank the reviewer for the comments and suggestions. We have revised the 2.1 Cohort recruitment subsection to improve clarity. The section now reads on lines 84-87: “To compare the nasal mycobiome of infants born within one year after Hurricane Maria to those born one to two years after the hurricane, we recruited infants between March and August 2018 (Year 1 Group) and then between February and September 2019 (Year 2 Group) at the San Juan City Hospital Research Unit.”
Comment 2: The manuscript is otherwise easy to read and follow. However, the authors do not attempt to establish associations between the nasal mycobiome and important respiratory outcomes, which would certainly be feasible, since at the age of 6-7 (at which the children should be in the present day) it would be possible to determine at least if these children are atopic/allergic, and have/have had recurrent wheezing/asthma. This would greatly improve the quality and reach of the manuscript.
Response 2: We agree that evaluating respiratory outcomes of our cohort would be highly valuable to establish the associations between observed dysbiosis in the mycobiome and childhood asthma/wheezing. Unfortunately, we did not follow the participants to assess their respiratory outcomes in this study. Therefore, we acknowledged this limitation and advocated for future studies in this area to include respiratory health outcomes on lines 275-279: “Third, we did not follow the participants to assess their respiratory outcomes, such as wheezing, viral respiratory infections, and childhood asthma. Studies that integrate airway mycobiome profiling and respiratory health outcomes are needed to further evaluate the associations between nasal fungal dysbiosis during infancy and the risks of developing respiratory diseases.”
Comment 3: Line 74: Aims should be in a new paragraph.
Response 3: Thank you for this comment. We have revised this accordingly.
Comment 4: Lines 130-137: this sentenced is redundant and does not add information besides what is already clear in table 1, and should be removed.
Response 4: Thank you for this comment. We have removed the detailed description of Table 1.
Comment 5: Line 253-4: do the authors mean have been related to respiratory diseases?
Response 5: Thank you for this clarification. Yes, we have updated the language to “respiratory disease.”
Comment 6: Line 256: Schizophyllum should be in italic
Response 6: Thank you for this correction. We have revised this accordingly.
Comment 7: Limitations should include the absence of a control group from a non-flooded area.
Response 7: Thank you for this important comment. Hurricane Maria impacted almost the entire island of Puerto Rico. Therefore, we could not identify a non-flooded area within the island as a control group. In addition, recruiting a control group outside of Puerto Rico may have introduced a significant bias due to environmental and meteorological variations between two different catchment regions. However, we recognized that the control group (Year 2 Group) may also have been indirectly affected by the hurricane. This potential limitation has now been acknowledged on lines 279-283: “Fourth, we used a cohort of infants who were born between one and two years after Hurricane Maria as the control group to identify hurricane-related dysbiosis. It is important to note, however, that the profound impacts of the devastating Hurricane Maria were long-lasting and could have affected the control group of infants as well.”
Reviewer 2 Report
Comments and Suggestions for Authors
I have some comments and suggestions for this manuscript as follows:
- Clearly state in the table legend or text that maternal mold exposure data was only collected/available for Year 1 Group mothers (n=8 reported). The dash for Year 2 is ambiguous. Explicitly note if Year 2 mothers were asked and reported no exposure or if this data wasn't collected.
- The abstract highlights Alternaria, Eutypella, Schizophyllum, and Auricularia. Provide a brief rationale in the results or discussion for specifically highlighting these four DESeq2-identified genera for abundance/prevalence comparisons, beyond just statistical significance, especially given other genera were also differentially abundant.
- The reference to "Scheme 2" in the Table 1 footnote appears erroneous (likely meant "Year 2 Group"). Correct this term to "Year 2 Group" for clarity and consistency with the rest of the manuscript.
- Ensure the caption for Figure S1 clearly states that these analyses (evaluating age, sex, breastfeeding, siblings) were performed across the entire cohort (Year 1 + Year 2) to assess potential confounding factors, not just within specific groups. This context is crucial.
- While Alternaria's association with asthma is cited, directly acknowledge the study's limitation in not measuring respiratory outcomes. Briefly suggest how the observed dysbiosis specifically might mechanistically contribute to increased asthma risk in the discussion, beyond mere association with exposure.
- Some recent studies indicate that early-life exposure to indoor mold/damp stains (as an important type of fungal spore) are associated with childhood asthma. Furthermore, specific climatic factors such as temperature and relative humidity (closely related to indoor fungal spore) exposure during early life are also indicated to be associated with childhood asthma and allergies. Please compare and discuss your findings with these new articles to further support your results:
[1] https://doi.org/10.1016/j.buildenv.2022.109740
[2] https://doi.org/10.1016/j.buildenv.2023.110668
Author Response
Comment 1: Clearly state in the table legend or text that maternal mold exposure data was only collected/available for Year 1 Group mothers (n=8 reported). The dash for Year 2 is ambiguous. Explicitly note if Year 2 mothers were asked and reported no exposure or if this data wasn't collected.
Response 1: We appreciate the reviewer’s comments and suggestions. Mothers in the Year 2 Group were asked about Hurricane Maria-related mold exposure during their pregnancy, and all reported no such exposure. We have revised Table 1 accordingly and clarified this point in the text.
Comment 2: The abstract highlights Alternaria, Eutypella, Schizophyllum, and Auricularia. Provide a brief rationale in the results or discussion for specifically highlighting these four DESeq2-identified genera for abundance/prevalence comparisons, beyond just statistical significance, especially given other genera were also differentially abundant.
Response 2: Thank you for this suggestion. We had highlighted these four fungal genera – Alternaria, Eutypella, Schizophyllum, and Auricularia – as they were identified to be more abundant in infants from the Year 1 Group compared to the Year 2 Group, and these are also closely related to respiratory health. We have emphasized the importance of these four genera in the Discussion section. While four other genera were also identified to be more abundant in the Year 2 Group than the Year 1 Group, they were not related to hurricane exposures or lung health. Therefore, due to space limitations, we did not mention them in the abstract. However, they are discussed in the Results and Discussion sections.
Comment 3: The reference to "Scheme 2" in the Table 1 footnote appears erroneous (likely meant "Year 2 Group"). Correct this term to "Year 2 Group" for clarity and consistency with the rest of the manuscript.
Response 3: Thank you for this correction. This sentence has been removed in the Table 1 footnote.
Comment 4: Ensure the caption for Figure S1 clearly states that these analyses (evaluating age, sex, breastfeeding, siblings) were performed across the entire cohort (Year 1 + Year 2) to assess potential confounding factors, not just within specific groups. This context is crucial.
Response 4: Thank you for this suggestion. We have clarified this in the caption for Figure S1 accordingly.
Comment 5: While Alternaria's association with asthma is cited, directly acknowledge the study's limitation in not measuring respiratory outcomes. Briefly suggest how the observed dysbiosis specifically might mechanistically contribute to increased asthma risk in the discussion, beyond mere association with exposure.
Response 5: Thank you for this comment. We have added a sentence in the Discussion section on lines 242-244 to describe the mechanisms by which Alternaria may trigger asthma on line 242-244: “Notably, Alternaria exposures have been closely associated with childhood asthma development and severe asthma exacerbations through airway epithelial barrier disruption and immune activation 12, 27−30.”
Comment 6: Some recent studies indicate that early-life exposure to indoor mold/damp stains (as an important type of fungal spore) are associated with childhood asthma. Furthermore, specific climatic factors such as temperature and relative humidity (closely related to indoor fungal spore) exposure during early life are also indicated to be associated with childhood asthma and allergies. Please compare and discuss your findings with these new articles to further support your results:
[1] https://doi.org/10.1016/j.buildenv.2022.109740
[2] https://doi.org/10.1016/j.buildenv.2023.110668
Response 6: Thank you for the comments and suggestions. We have added a few sentences in the Discussion section on lines 269-274 accordingly: “Second, prior studies have shown that exposure to indoor mold and dampness during the prenatal stage is associated with increased risks of developing asthma 35. It is possible that prenatal hurricane exposures played a role in mycobiome dysbiosis in the Year 1 Group since all the infants in that group were exposed in utero to Hurricane Maria. However, given the study design, we cannot distinguish the prenatal and postnatal effects on the infant nasal mycobiome.”
Round 2
Reviewer 2 Report
Comments and Suggestions for Authors
Many thanks for the authors’ careful explanations and revision. The authors have addressed all my questionnaires and comments. Now, I have no further comments.